# Exploring the Rumen Microbiota and Serum Metabolite Profile of Hainan Black Goats with Different Body Weights before Weaning

**DOI:** 10.3390/ani14030425

**Published:** 2024-01-28

**Authors:** Gang Zheng, Dongxing Wang, Kai Mao, Musen Wang, Jian Wang, Wenjuan Xun, Shuai Huang

**Affiliations:** Forage Processing and Ruminant Nutrition Laboratory, School of Tropical Agriculture and Forestry, Hainan University, Haikou 570228, China; zhenggang97120815@163.com (G.Z.);

**Keywords:** goat kids, body weight, rumen microbiota, serum metabolome

## Abstract

**Simple Summary:**

Low body weight is not conducive to the growth and development of mammals in the subsequent stages of weaning. The rumen microbiota plays an important role in the growth, development, production, and health of ruminants. A comprehensive analysis was conducted using a combined 16S rRNA and metabolomics analysis method to determine the impact of the rumen microbiota and serum metabolites on goat kids before weaning weight. This study may help to provide new insights for further regulating the rumen microbiota to improve the body weight of animals.

**Abstract:**

The critical role of the rumen microbiota in the growth performance of livestock is recognized, yet its significance in determining the body weight of goat kids before weaning remains less understood. To bridge this gap, our study delved into the rumen microbiota, serum metabolome, rumen fermentation, and rumen development in goat kids with contrasting body weights before weaning. We selected 10 goat kids from a cohort of 100, categorized into low body weight (LBW, 5.56 ± 0.98 kg) and high body weight (HBW, 9.51 ± 1.01 kg) groups. The study involved sampling rumen contents, tissues, and serum from these animals. Our findings showed that the HBW goat kids showed significant enrichment of VFA-producing bacteria, particularly microbiota taxa within the *Prevotellaceae* genera (*UCG-001*, *UCG-003*, and *UCG-004*) and the *Prevotella* genus. This enrichment correlated with elevated acetate and butyrate levels, positively influencing rumen papillae development. Additionally, it was associated with elevated serum levels of glucose, total cholesterol, and triglycerides. The serum metabonomic analysis revealed marked differences in fatty acid metabolism between the LBW and HBW groups, particularly in encompassing oleic acid and both long-chain saturated and polyunsaturated fatty acids. Further correlational analysis underscored a significant positive association between *Prevotellaceae_UCG-001* and specific lipids, such as phosphatidylcholine (PC) (22:5/18:3) and PC (20:3/20:1) (r > 0.60, *p* < 0.05). In summary, this study underscores the pivotal role of the rumen microbiota in goat kids’ weight and its correlation with specific serum metabolites. These insights could pave the way for innovative strategies aimed at improving animal body weight through targeted modulation of the rumen microbiota.

## 1. Introduction

Body weight before weaning influences the growth of suckling animals and further affects the performance of animals in subsequent stages [1]. Previous studies on piglets have shown that low weaning weight may negatively impact lifetime growth rate [2], meat character [3], feed intake, and carcass traits [4]. Studies have demonstrated that low-weight piglets reduce their feed intake compared to normal-weight piglets, and their digestive and immune systems were incompletely developed after weaning [5]. These conditions increase the incidence of various diseases and microbiota infections due to the vulnerability of low-weight piglets after weaning. Further research indicates that low-weight mice exhibit imbalanced lipid metabolism and insulin resistance in adulthood [6,7]. Hence, the low body weight of animals in their infancy is not conducive to their health, growth, and development.

The rumen harbors a highly diverse microbiota community contributing to powerful functions, including bacteria, archaea, fungi, and ciliates. This indicates that volatile fatty acids (VFAs), microbiota crude protein, and B-group vitamins are synthesized through the fermentation and transformation of feed by the rumen microbiota, serving as essential nutrients for ruminants [8]. Lopes et al. [9] found that Nellore cattle with high feed efficiency exhibited a relatively high abundance of *Anaerotruncus*, *Phocaeicola*, unclassified *Prevotella*, and lower levels of Bacteroidetes during the growth stage. High-growth performance Yak calves [10] and lambs [11] with milk replacer feeding were highly enriched in the *Prevotella* and *Christenselelaceae_R-7_group*. The enrichment of *Sediminibacterium* and *Butyrivibrio* in the intestine can promote weight gain in weaned piglets, as shown in previous studies [12]. Another study on meat rabbits found that *Blautia*, *Lachnoclostridium*, and *Butyricoccus* with high relative abundance in feces were not conducive to weaning weight gain [13]. These studies mainly address monogastric animals, and the impact of the rumen microbiota on body weight in ruminant animals before weaning is not clear.

The objective of this study was to investigate the rumen microbiota taxa based on varying body weights and identify specific associations with serum metabolites, intensively investigating the intricate interactions between the microbiota and host. Hence, we hypothesized that within a consistent feeding environment, the extent of rumen fermentation and development is intricately linked to variations in the rumen microbiota and that there exists a specific correlation with serum metabolites. This research offers novel insights and serves as a valuable reference for elucidating the potential impact of the rumen microbiota on body weight, and it may contribute to strategies aimed at enhancing animal weight through microbiota modulation.

## 2. Materials and Methods

### 2.1. Experimental Design, Animals, and Sample Collection

The experiment strictly adhered to Hainan University’s animal research guidelines in Haikou, China, and received approval from the institution’s Animal Protection Committee under approval number (HNUAUCC-2024-00002). We utilized 100 Hainan black goat buck kids, all 21 days old with comparable birth weights, for this study. These goat kids were housed in individual pens until they reached 90 days of age, as detailed in Table 1. Throughout the experiment, biweekly weight measurements were conducted from birth until day 90. The goat kids were delimited into two test groups according to their body weight at 90 days. Those weighing above the average for 90-day-old goat kids were placed in the high body weight group (HBW, averaging 9.51 ± 1.01 kg). In contrast, those below the average were assigned to the low body weight group (LBW, averaging 5.56 ± 0.98 kg). Each group of goat kids adopts a simple random sampling method, with 5 biological replicates in each group as representative samples. All goat kids were raised alongside their ewes for the first 21 days post-birth. Following this, there were separate feedings at 08:30 and 16:00 h of milk replacer (Hangzhou Xiangmu Fresh Biotechnology Co., Ltd., Hangzhou, China) twice per day (totaling 1 L/d) from days 21 to 90. Additionally, from the age of 45 days, the goat kids were fed a mixed diet that gradually increased by 5 g per day (starting from 5 g and reaching up to 220 g per day by the experiment’s end) to ensure the absence of residual feed; the experimental results are not biased by differences in dietary intake. This mixed feed comprised 58% concentrate and 42% roughage on a dry matter standard, as outlined in Appendix A. Access to fresh water was provided ad libitum. In the process of grouping, we ensured that there were no dramatic differences in birth weight and body weight at 21 days among the two experimental groups. This step was crucial to obviate the potential influence of forage and beginning weight before weaning body weight. The genetic background of the goat kids in both groups was similar, with them all being offspring of the same ram. To maintain the integrity of the experiment, the goat kids were not exposed to antibiotics at any point, and uniform conditions were maintained for all goat kids throughout the study.

After the experiment, blood samples were obtained from the jugular vein of all grouped goat kids after fasting for 12 h, and they were then euthanized and dissected for sampling. After centrifugation for 10 min at 3100 rpm/min, the serum was divided into two parts. Chemical indicators were measured and stored at −20 °C, while the other part was analyzed for metabolomics and preserved at −80 °C.

We measured the pH value of the rumen after dissection using a pH meter (DDS-307, Geomagnetic Company, Shanghai, China). The rumen contents samples used for DNA extraction were stored in liquid nitrogen at −80 °C. The rumen fluid sample obtained through the use of a filtering cheesecloth with four layers was preserved at −20 °C for follow-up analysis and determination. The rumen of the individuals was weighed after it was emptied, washed with tap water, and dried with a paper towel.

The rumen epithelial samples (1.0 cm^2^) collected from the experimental goat kids were rinsed with sterile phosphate-buffered saline (PBS). The rumen tissue used for the tissue morphology analysis was preserved in 4% paraformaldehyde.

### 2.2. Hematoxylin and Eosin (H&E) Staining

We evaluated the rumen morphology as described previously [14]. Briefly, the rumen tissue was washed with PBS, dehydrated with ethanol, embedded in paraffin, sliced, and finally stained with H&E (Appendix A). We measured the width, length, and muscle layer thickness of the rumen papillae using IXplore Pro microscope photography (Olympus China Group Co., Ltd., Shanghai, China).

### 2.3. Rumen Fermentation Parameters

A gas chromatography system (Clarus 680, PerkinElmer, Inc, Waltham, MA, USA) equipped with an Elite-FFAP column (0.25 mm i.d.; 30 m in length) was used to determine the VFA content in the rumen as per the guidelines of Chen et al. [15]. The colorimetric method was used to determine the ammonia nitrogen (NH_3_-N) content in the rumen fluid [16].

### 2.4. DNA Extraction, 16S rRNA Sequencing, and 16S rRNA Data Processing

In this study, we extracted the total genomic DNA from all samples by utilizing the QIAamp DNA Stool Mini Kit (Qiagen, Hilden, Germany). The concentration and purity of the abstracted genomic deoxyribonucleic acid were rigorously assessed using a NanoDrop 2000 spectrophotometer (Thermo Scientific, China Group Co, Ltd., Guangzhou, China). The bacterial 16S rRNA gene’s V3-V4 hypervariable region was selectively amplified by utilizing the universal primers 338F (5′-ACTCCTACGGGAGGCAGCAG-3′) and 806R (5′-GGACTACHVGGGTWTCTAAT-3′), including a 6-base pair error-correcting barcode, facilitating distinct identification of each sample in subsequent analyses. We used the Phusion High-Fidelity Polymerase Chain Reaction (PCR) Mastermix (New England Biolabs Beijing Ltd., Beijing, China). The repeating parameters were set as follows: an initial denaturation step for 5 min at 95 °C, followed by 28 repeating cycles running at 95 °C for 45 s, 55 °C for 50 s, and 72 °C for 45 s. This was concluded with a final extension phase of 72 °C for 10 min, ensuring thorough amplification of the target DNA sequences. The PCR products underwent quantification using 1% agarose gel electrophoresis, followed by a purification process utilizing the Agencourt AMPure XP Kit (Beckman Colter Genomics, Indianapolis, IN, USA). Quantification of amplicon libraries using a Qubit 2.0 fluorometer (Thermo Fisher Scientific, Waltham, MA, USA) was performed on all amplicons obtained from the samples. These libraries were then sequenced on a MiSeq PE300 platform (Illumina, San Diego, CA, USA), a procedure that generated paired-end reads of 2 × 250 base pairs each [17].

Sequence processing involving demultiplexing and quality filtering via QIIME [18] (http://qiime.org, accessed on 7 May 2023), retaining only bases with a quality score exceeding 20 and reads longer than 200 bp. FLASH [19], was utilized for merging paired-end reads into tags, necessitating a minimum 10-base sequence overlap. Chimeric sequences were identified and eliminated using UCHIME [20]. Operational taxonomic units (OTUs) [21] were clustered at a 97% similarity threshold with UPARSE2 [22] and classified taxonomically using the Ribosomal Database Project classifier [23] against the SILVA 138.1 database [24]. After the post-removal of singleton and doubleton OTUs via UCLUST [25], the representative OTU table was finalized. For the analyses of α-diversity, β-diversity, and taxonomic classification, normalized OTU counts per sample were examined using R 3.6.0 [26].

### 2.5. Serum Biochemistry Indices

The serum levels of glucose, blood urea nitrogen (BUN), alanine transaminase (ALT), aspartate transaminase (AST), alkaline phosphatase (ALP), lactic dehydrogenase (LDH), total cholesterol (TC), triglycerides (TG), and protein were determined using a Cobus-Mira-Plus automatic clinical chemistry analyzer (Cobus-Mira-Plus; Roche Diagnostic Systems Inc., Zurich, Swiss Confederation). The serum globulin content was computed by deducting the albumin level from the total protein.

### 2.6. Serum Metabolome

In this study, we employed a cutting-edge approach to analyze the serum metabolome, leveraging the precision of a UHPLC system [27] (Vanquish, Thermo Fisher Scientific) integrated with a Q Exactive HFX mass spectrometer (Orbitrap MS, Thermo). We initiated our protocol by treating 100 µL of serum with 300 µL of a methanol–acetonitrile mixture. The samples underwent a thorough vortex process, followed by two-hour incubation at −20 °C. Post-incubation, they were centrifuged, freeze-dried, and redissolved.

The serum samples’ intricate analysis was executed using ProteoWizard for data conversion and a bespoke R-based program, harnessing XCMS for peak analysis. Our identification strategy hinged on matching both the mass spectrum and retention index. We adopted a stringent criterion, discarding metabolite peaks found in less than half of our samples. To address missing data, we filled gaps with half the minimum value, ensuring a comprehensive analysis. Altogether, 570 serum metabolites were characterized, so that data normalization processing uses the sum of features of each sample for downstream analysis.

In our study, we employed the “cluster” package (v3.6.0) in R for hierarchical clustering of pre-treatment serum metabolome samples, applying orthogonal partial least-squares discriminant analysis (OPLS-DA). We pinpointed differential metabolites based on a two-fold criterion: a variable importance in projection (VIP) score exceeding one from the OPLS-DA model and a *p*-value below 0.05 from the univariate statistical analysis. Subsequently, we conducted a comprehensive functional analysis of these metabolites using MetaboAnalyst 5.0 (https://www.metaboanalyst.ca, accessed on 24 May 2023) [28], which provided valuable insights into their biological roles and potential impact.

### 2.7. Correlation Analysis

Utilizing the “vegan” R package (v3.6.0), we executed a Procrustes test to assess the alignment of two-dimensional shapes derived from PCoA analyses across two datasets. We applied Spearman correlation analysis to investigate relationships among distinct bacteria, rumen fermentation parameters, rumen morphology, serum biochemical indices, and serum metabolites. Emphasis was placed on significant correlations, defined by a threshold of an absolute Spearman rank-order correlation coefficient (|r|) greater than 0.6 and a *p*-value less than 0.05. These significant correlations were then effectively visualized using the “pheatmap” package in R.

### 2.8. Statistical Analysis

We utilized R version 4.2.3 for analyzing body weight, serum parameters, rumen morphology, rumen fermentation parameters, rumen microbiota, and serum metabolites by employing a linear mixed-effects model:$$Y_{ik}=\mu +O_{i}+L_{K}+e_{ik}$$

The dependent variable is represented as $Y_{ik}$, and the overall average is represented as μ. $O_{i}$ is the effect of the group, $L_{k}$ is the random effect of goat kids, and $e_{ik}$ is the residual error.

For the analysis of body weight, rumen morphology, serum biochemical indices, and rumen fermentation parameters across different groups, we conducted independent *t*-tests using SPSS 21.0 (https://www.ibm.com/cn-zh/spss, accessed on 12 August 2023). The alpha diversity indices of the rumen microbiota were assessed using the Wilcoxon rank-sum test, considering *p* < 0.05 as the threshold for statistical significance. Additionally, permutational multivariate analysis of variance (PERMANOVA) was performed using the vegan package in R, focusing on genus-level bacterial taxa.

We compared the relative abundance of genera (with relative abundances exceeding 0.10% in at least 70% of samples) using the Wilcoxon rank-sum test. *p*-values were adjusted for false discovery rate, with a threshold of *p* < 0.05 after correction deemed significant for our study.

## 3. Results

### 3.1. Rumen Fermentation Parameters of Goat Kids

In our findings, significant differences were observed in rumen NH_3_-N levels between the HBW and LBW groups (*p* < 0.05, Table 2). Furthermore, goat kids with HBW exhibited a significantly elevated concentration of total VFA, encompassing acetate, butyrate, and valerate, in comparison to goat kids with LBW, a statistically significant difference (*p* < 0.05).

### 3.2. The Difference in the Rumen Morphology of the Goat Kids

The study revealed that the length and width of the rumen papillae in the HBW group are more developed than those in the LBW group, with statistical significance (*p* < 0.05, Appendix A). Nevertheless, the groups characterized by HBW and LBW demonstrated comparable developmental progress in aspects such as rumen weight and muscle thickness (*p* > 0.05, Appendix A).

### 3.3. Serum Biochemical Indicators of the Goat Kids

Serum biochemical analysis revealed elevated levels of key nutrients, including glucose, TC, and TG, in the HBW group when compared to the LBW group (*p* < 0.05, Appendix A).

### 3.4. Rumen Microbiota Sequencing and Composition

From the ten samples analyzed, we initially obtained 539,364 raw bacterial sequences. Following quality control and equal-depth clustering with 97% identical sequences (53,936 reads per sample), the study identified a total of 7067 operational taxonomic units (OTUs). These OTUs were classified into 25 phyla, 48 classes, 84 orders, 139 families, and 298 genera. Rarefaction curves, as shown in Appendix A, confirmed that the sampling depth adequately represented the rumen microbiota, evidenced by a plateau in the number of new OTUs despite increasing sequence counts per sample. The Good’s coverage exceeded 99.03%, further supporting the sufficiency of our sequencing depth for this microbiota study.

Bacteroidetes, Firmicutes, and Proteobacteria emerged as advantageous groups at the phylum level (Appendix A). The most prevalent genera included *F082*, *Bacteroides*, *Rikenellaceae_RC9_gut_group*, *Prevotella*, *Ruminococcus*, *Muribaculaceae*, and *Oscillospiraceae_UCG-005* (Appendix A).

### 3.5. Comparison of the Rumen Microbiota between Goat Kids with Different Body Weights

In assessing microbiota α-diversity using the number of OTUs, the Chao1 index, and the Shannon index through Wilcoxon rank-sum tests, significant differences emerged between the HBW and LBW groups. Specifically, the HBW group exhibited notably higher values in these indices (*p* < 0.05, Appendix A), suggesting reduced α-diversity in LBW goat kids compared to their HBW counterparts.

To determine the profile of the rumen microbiota communities in the LBW and HBW groups, β-diversity was assessed using the Procrustes test and visualized using a PCoA plot. As shown in the PCoA plot, the rumen samples of the HBW and LBW groups cross each other and overlap (Figure 1). To further compare the significant difference in the rumen microbiota composition between the tested groups, PERMANOVA analysis was used. PERMANOVA analysis showed that there was a significant difference in microbiota composition between the HBW and LBW groups (*p* = 0.028).

At the phylum and genus levels, assessed via Wilcoxon rank-sum tests, we observed significant disparities in microbiota abundance between the groups. In HBW goat kids, there was a significantly higher relative abundance of Bacteroidetes (*p* = 0.011) and a significantly lower relative abundance of Firmicutes (*p* = 0.05, Table 3). Other dominant phyla like Proteobacteria, Desulfobacterota, and Spirochaetota showed no significant differences (*p* > 0.05, Table 3).

Significant differences were observed at the genus level between the HBW and LBW groups. In the HBW group, genera such as *F082*, *Rikenellaceae_RC9_gut_group*, *Prevotella*, and several *Prevotellaceae* subgroups (*UCG-001*, *UCG-003*, *UCG-004*), along with *Syntrophococcus*, *Lachnospiraceae_UCG-002*, *Pseudobutyrivibrio*, *Ruminococcus*, *Veillonellaceae_UCG-001*, and Eubacterium_nodatum_group, showed significantly higher relative abundance compared to the LBW group (*p* < 0.05, Table 4). Conversely, the abundance of *Muribaculaceae*, *Escherichia-Shigella*, *Bacteroides*, *Alistipes*, *Butyricimonas*, *Ruminococcus_torques_group*, and *Oscillospiraceae_UCG-005* was significantly lower in the HBW group compared to the LBW group (*p* < 0.05, Table 4). This indicates a distinct microbiota composition associated with body weight variations.

### 3.6. Correlation Analysis of Rumen Microbiota and Rumen Fermentation Parameters, Rumen Morphology, and Serum Biochemical Indicators

The Spearman correlation result demonstrated that acetate and butyrate were significantly and positively correlated with the relative abundance of *Prevotellaceae_UCG-003*, *Prevotellaceae_UCG-001*, *Prevotella*, and *F082* (r > 0.60, *p* < 0.05, Figure 2). The rumen papillae length and width were significantly and positively correlated with the relative abundance of *Rikenellaceae_RC9_gut_group*, *Prevotellaceae_UCG-001,* and *Pseudobutyrivibrio* (r > 0.60, *p* < 0.05, Figure 2). Glucose, TC, and TG were significantly and positively correlated to the relative abundance of *Prevotellaceae_UCG-001*, *Prevotellaceae_UCG-003*, *Rikenellaceae_RC9_gut_group*, and *Pseudobutyrivibrio* (r > 0.60, *p* < 0.05, Figure 2). The relative abundance of F082, *Ruminococcus*, and *Veillonellaceae_UCG-001* was positively correlated with pH, while the relative abundance of *Alistipes*, *Butyricimonas*, and *Oscillospiraceae_UCG-005* was negatively correlated with pH (r < −0.60, *p* < 0.05, Figure 2). In addition, the enrichment of *Pseudobutyrivibrio* was significantly and positively correlated with NH_3_-N (r > 0.60, *p* < 0.05, Figure 2).

### 3.7. Differences in the Serum Metabolomics and Metabolic Pathways of Goat Kids with Different Weaning Weights

To investigate metabolic changes between the two groups, we conducted LC-MS analysis on serum metabolites. The OPLS-DA models showed excellent interpretative and predictive abilities, with R^2^Y and Q^2^ values of 0.999 and 0.417 in both positive and negative ion modes (Appendix A). A distinct separation in the OPLS-DA plots between the HBW and LBW groups was observed, indicating differences in their metabolic profiles (Appendix A).

In the present study, we identified 570 metabolites. Notably, 20 metabolites, including diacetone alcohol, citraconic acid, 16-hydroxy hexadecanoic acid, cis,cis-muconic acid, 3-guanidinopropionate, 2-hydroxybutyric acid, methylimidazole acetaldehyde, bovinic acid, myristoleic acid, 2-hydroxymyristic acid, adipic acid, homocitrulline, gamma-aminobutyric acid, alpha-linolenic acid, PC (20:3/20:1), and PC (22:5/18:3), were found in higher concentrations in the HBW group. Conversely, lsoliquiritgenin, 2-methylpiperidine, and two other phosphatidylcholines were more abundant in the LBW group (*p* < 0.05, Figure 3A). KEGG pathway analysis revealed that the above differential metabolites were mainly concentrated in “valine, leucine, and isoleucine biosynthesis”, “linoleic acid metabolism”, “sphingolipid metabolism”, “biosynthesis of unsaturated fatty acids”, “propanoate metabolism”, and “alpha-linolenic acid metabolism” pathway (pathway impact > 0.1, *p* < 0.05, Figure 3B). This highlights a pronounced metabolic divergence between HBW and LBW goat kids.

### 3.8. Correlation between the Major Rumen Microbiota and Different Serum Metabolites

To elucidate the potential interplay between the rumen microbiota and serum metabolism, we employed the Spearman correlation results, which examined the associations between the rumen microbiota and serum metabolites (Figure 4). The analysis revealed notable correlations. Specifically, the relative abundance of *Defluviitaleaceae_UCG-011* exhibited a positive correlation with certain lipids and lipid-like molecules. These include bovinic acid (r = 0.82, *p* = 0.01), myristoleic acid (r = 0.72, *p* = 0.04), citraconic acid (r = 0.83, *p* = 0.01), and 2-hydroxymyristic acid (r = 0.80, *p* = 0.01).

Additionally, *Prevotellaceae_UCG-001* was found to be positively correlated with four lipid metabolites, namely adipic acid (r = 0.81, *p* = 0.01), PC (20:3/20:1) (r = 0.80, *p* = 0.01), PC (18:2/15:0) (r = 0.83, *p* = 0.01), and cis, cis-muconic acid (r = 0.76, *p* = 0.03). Furthermore, the relative concentration of PC (18:2/15:0) showed a positive correlation with the abundance of *Bacteroides* (r = 0.81, *p* = 0.01) and the *Ruminococcus_torques_group* (r = 0.82, *p* = 0.01).

The research also identified a positive correlation between the relative concentration of PC (18:2/14:0) and the abundance of *Butyricimonas* (r = 0.67, *p* = 0.03). Conversely, a negative correlation was observed between the relative concentration of 2-hydroxymyristic acid and the abundance of *Bacteroides* (r = −0.66, *p* = 0.03) and *Oscillospiraceae_UCG-005* (r = −0.61, *p* = 0.04). These findings provide insights into the complex relationships between the rumen microbiota and host metabolic processes.

## 4. Discussion

In this research, we detail the rumen microbiota profiles in LBW and HBW goat kids, identifying significant links between the rumen microbiota, growth performance, and host metabolism. Our findings revealed that (i) the rumen microbiota in HBW goat kids was predominantly composed of bacteria adept at VFA production and (ii) a strong correlation exists between the rumen fermentation parameters, serum glucose, and rumen microbiota composition.

VFAs, a crucial energy source for ruminants, significantly contribute to their metabolic energy, accounting for up to 50% of their energy intake [29,30]. Research by Górka et al. [31,32] demonstrated that sodium butyrate enhances rumen development and expedites the growth of the rumen papilla in calves. Additionally, intraluminal infusion of acetate has been found to stimulate rumen development in non-lactating cows [33]. In our research, the HBW goat kids exhibited superior levels of butyrate and acetate, along with increased rumen papilla length and width, compared to the LBW goat kids. This suggests that HBW goat kids possess a more advanced rumen fermentation capacity, which in turn enhances the development of rumen papillae.

In the current study, we observed a notable disparity in the microbiota composition at both the phylum and genus levels between the LBW and HBW goat kids, aligning with the findings of Ding et al. [12] and Fang et al. [13]. Notably, the genus Ruminococcus was significantly more abundant in HBW goat kids, corroborating the findings of Wang et al. [34]. As identified by Gaffney et al. [35], *Ruminococcus* is known for its butyrate-producing capabilities. Additionally, we detected a significantly higher relative abundance of *Prevotellaceae_UCG-001*, *Prevote-llaceae_UCG-004*, *Prevotellaceae_UCG-003*, and *Prevotella* in the HBW goat kids. These bacteria, as detailed by Seshadri et al. [36] and Accetto et al. [37], are adept at polysaccharide degradation and produce acetate and butyrate. Previous research, such as the work by Chiquette et al. [38], demonstrated that the introduction of *Prevotella bryantii 25A* into the rumen of dairy cows via a rumen fistula led to increased butyrate levels. Similarly, Takizawa et al. [39] found that microbiota translocation in Japanese black cattle resulted in a higher abundance of *Prevotellaceae_UCG-004*, subsequently elevating acetate and butyrate levels. This was further supported by Zhao et al. [40], who confirmed through in vitro fermentation tests that *Prevotellaceae_UCG-001* contributes to lignocellulose degradation and elevates VFA concentrations. Moreover, our study revealed a positive correlation between the relative abundance of *Prevotella*, *Prevotellaceae_UCG-003*, and butyrate levels, echoing the findings of Wang et al. [41] in young goats, who also reported a positive correlation between the abundance of these bacteria and the concentrations of acetate and butyrate. These findings collectively suggest that the enriched presence of VFA-producing bacteria, including *Prevotellaceae_UCG-001*, *Prevotellaceae_UCG-003*, *Prevotellaceae_UCG-004*, and *Prevotella*, in the HBW goat kids is instrumental in enhancing rumen fermentation ability and the development of the rumen epithelium.

Glucose is a crucial nutrient for ruminants, with it playing a pivotal role in their metabolic processes. According to Leskova et al. [42], serum glucose levels in ruminants are regulated by propionate levels, a primary substrate for hepatic gluconeogenesis. In our investigation, we observed higher concentrations of VFA and propionate in the rumen of the HBW group compared to the LBW group. Correspondingly, the HBW goat kids exhibited significantly higher serum levels of glucose, TC, and TG when compared with the LBW group. Recent research by Xu et al. [43] aligns with our findings, demonstrating that rumen microbiota enhancement leads to increased production of propionate and VFAs. This, in turn, elevates serum glucose levels, subsequently boosting the average daily weight gain in Holstein heifer calves pre-weaning. Furthermore, elevated serum glucose levels have been implicated in increasing substrate concentrations across various metabolic pathways. This enhances the synthesis of TG and TC in serum, factors integral to growth, development, and metabolism, as supported by studies from Ma et al. [44] and Schade et al. [45]. A noteworthy aspect of our study is the Spearman analysis, which revealed a simultaneous and positive correlation between glucose, TG, and TC and the relative abundance of specific rumen bacteria, namely *Prevotellaceae_UCG-001* and *Prevotellaceae_UCG-004*. These findings suggest a potential mechanistic link whereby higher VFA-producing bacteria in the rumen may elevate serum glucose levels, contributing significantly to the growth of the animals.

Lipid metabolism plays a critical role in body weight regulation. Garcia et al. [46] previously demonstrated that an increase in linoleic acid and α-linolenic acid levels in the blood before weaning can enhance the growth of Holstein calves. In our study, we observed significant alterations in fatty acid profiles, marked by a substantially higher concentration of various lipid metabolites in the serum of HBW goat kids compared to LBW goat kids. These metabolites include cholesterol esters, triglycerides, phospholipids, PC, acylcarnitine, sphingomyelins, and ceramides. Building upon previous research, Sherratt et al. [47] highlighted that PC is a vital lipid molecule for animal growth, influencing fatty acid oxidation and lipid transport. Furthermore, Liu et al. [48] found that dietary supplementation of PC in weaned piglets promoted lipid synthesis while inhibiting lipid catabolism. Consistent with these findings, our data revealed that in the HBW group, serum levels of specific PC species, namely PC (20:3/20:1) and PC (22:5/18:3), were significantly elevated and positively correlated with the abundance of *Prevotellaceae_UCG-001*. Interestingly, the work of Li et al. [49] supports the notion that the enrichment of *Prevotellaceae_UCG-001* in the intestines of mice, particularly those fed with Jerusalem artichoke and inulin, is closely associated with increased PC levels. Additionally, our study revealed that HBW goat kids exhibit elevated relative concentrations of adipic acid and 2-hydroxymyristic acid. These increases are positively correlated with the relative abundance of *Prevotellaceae_UCG-001*. Vlaeminck et al. [50] previously showed that *Prevotellaceae_UCG-001* plays a role in the hydrogenation of long-chain fatty acids and is strongly associated with odd and branched-chain fatty acids. Further supporting the role of *Prevotellaceae_UCG-001*, Ma et al. [51] demonstrated that its enrichment in the intestines of young rabbits leads to increased concentrations of short-chain fatty acids such as acetate, propionate, and butyrate. Similarly, Li et al. [52] found that the intestinal microbiota of mice enhances fatty acid synthesis by supplying a high level of acetate, a precursor for the synthesis of palmitate and stearate. In humans, Van der Beek et al. [53] observed that acetate infusion in the distal colon alters systemic lipid metabolism, impacting plasma triglyceride levels and body weight. Therefore, the growth rate of individual animals can be influenced by variations in their microbiota and metabolic profiles [54]. Drawing from these findings, we speculate that *Prevotellaceae_UCG-001* elevates VFA concentrations in the rumen, which in turn augments the lipid content in the serum of goat kids before weaning. This intricate interplay between the rumen microbiota and lipid metabolism offers valuable insights into the mechanisms governing early growth and development in goat kids.

## 5. Conclusions

Under the same feeding background, there were differences in the rumen microbiota and serum metabolites between the HBW and LBW goat kids, and there are metabolic interactions, suggesting that these differences may be important factors affecting body weight. Nevertheless, additional investigation is warranted to substantiate the speculation derived from these experimental results. The present study enhances our understanding of rumen microbiota in goat kids of different weights, providing insights for optimizing ruminant growth through targeted microbiota manipulation in animal husbandry practices.

## Figures and Tables

**Figure 1 animals-14-00425-f001:**
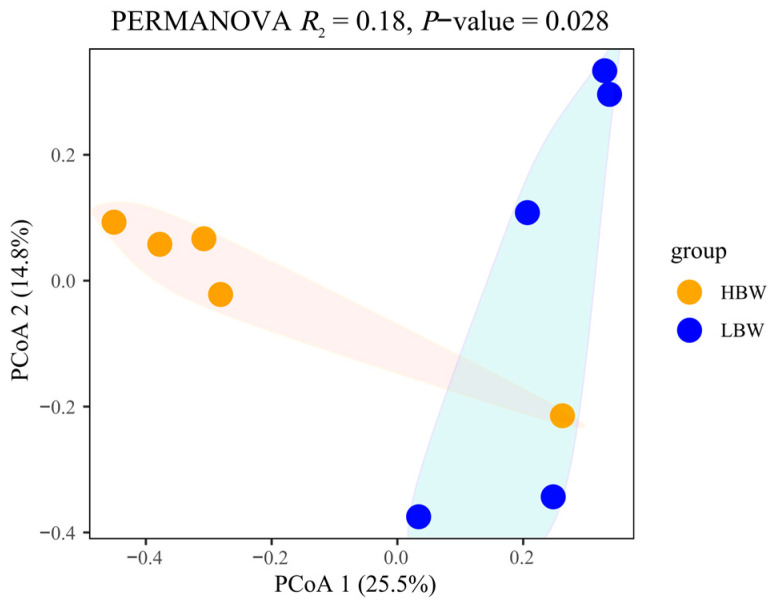
Comparison of the composition of the microbiota in the rumen of the goat kids. The PERMANOVA analysis with 999 permutations is shown. The microbiota composition of the HBW and LBW rumen samples based on OTUs was visualized using principal coordinate analysis (PCoA).

**Figure 2 animals-14-00425-f002:**
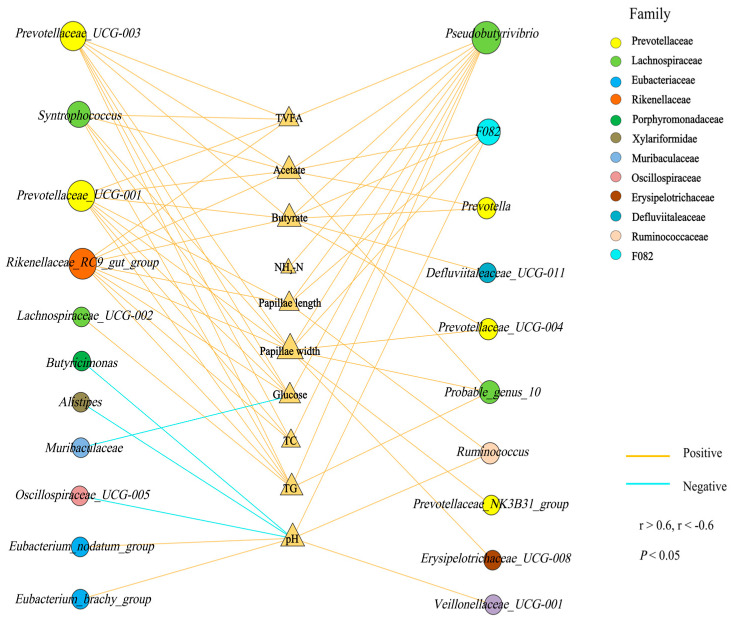
Correlation between the different genera and rumen fermentation parameters, rumen morphology, and serum biochemical indicators. Yellow represents a positive correlation, and blue represents a negative correlation. Only strong (Spearman r > 0.6, r < −0.6) and significant (*p* < 0.05) correlations are displayed.

**Figure 3 animals-14-00425-f003:**
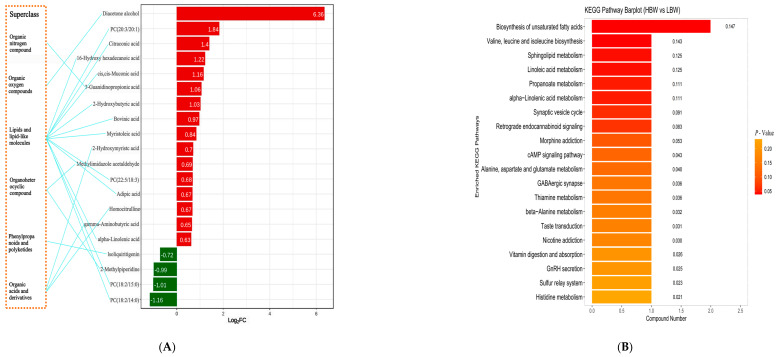
Serum metabolites of the HBW and LBW goat kids. (**A**) Error bars display significant metabolite differences in the serum between the HBW and LBW goat kids (red, HBW is significantly greater than LBW; green, HBW is significantly less than LBW). (**B**) Pathway enrichment analysis was performed using the significantly different serum metabolites between the HBW and LBW goat kids. Larger sizes, darker colors, and higher column numbers represent greater pathway enrichment, more significant enrichment, and higher pathway impact values, respectively.

**Figure 4 animals-14-00425-f004:**
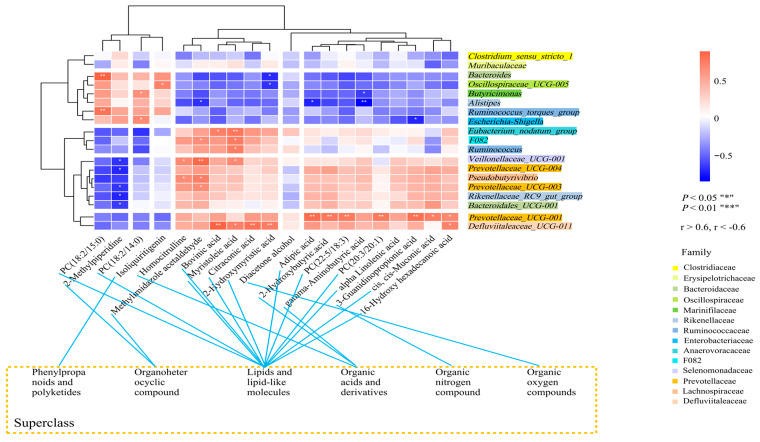
Interactions between the rumen microbiota and serum metabolites. Spearman correlations between significantly different rumen microbiota and serum metabolites (r > 0.6, r < −0.6; “*” represents *p* < 0.05 and “**” represents *p* < 0.01).

**Table 1 animals-14-00425-t001:** The birth weight and weaning weight of the LBW and HBW goat kids.

Items ^1^	Group ^2^		SEM ^3^	*p*-Value
	HBW	LBW		
Birth BW (kg)	2.33	2.17	0.22	0.417
21-day BW (kg)	3.73	3.65	0.19	0.614
Weaning BW (kg)	9.51	5.56	0.69	0.034

^1^ BW, body weight. ^2^ HBW = high body weight. LBW = low body weight. ^3^ SEM, standard error of the mean.

**Table 2 animals-14-00425-t002:** Rumen fermentation parameters between the HBW and LBW goat kids.

Items ^1^	Group ^2^		SEM ^3^	*p*-Value
	HBW	LBW		
pH	6.13	6.84	0.29	0.57
NH_3_-N (mg/dL)	4.79	3.42	0.23	0.01
Acetate (mmol/L)	21.37	13.54	5.81	0.02
Propionate (mmol/L)	8.13	6.48	2.25	0.23
Butyrate (mmol/L)	5.98	3.03	2.01	0.05
Isobutyrate (mmol/L)	0.82	0.60	0.05	0.14
Valerate (mmol/L)	0.72	0.39	0.15	0.03
Isovalerate (mmol/L)	1.32	1.05	0.18	0.53
AP (mmol/L)	2.63	2.08	0.27	0.12
TVFA ^4^ (mmol/L)	38.37	25.12	10.49	0.02

^1^ AP, the acetate-to-propionate ratio; TVFAs, total VFAs. ^2^ HBW = high body weight; LBW = low body weight. ^3^ SEM, standard error of the mean. ^4^ TVFA, total volatile fatty acids.

**Table 3 animals-14-00425-t003:** Comparison of the relative abundance of the rumen microbiota at the phylum level (relative abundances > 0.10% in at least 70% of samples) of weaning goat kids.

Phylum	Group ^1^		SEM ^2^	*p*-Value ^3^
	HBW	LBW		
Bacteroidetes	57.60	28.10	20.94	0.01
Firmicutes	23.36	38.07	12.47	0.05
Proteobacteria	12.78	28.10	16.08	0.13
Desulfobacterota	0.94	2.52	1.11	0.45
Spirochaetota	0.42	1.70	0.90	0.83

^1^ HBW = high body weight; LBW = low body weight. ^2^ SEM, standard error of the mean. ^3^ Adjusted *p*-value = false discovery rate-adjusted *p*-value; the *p*-values between the HBW and LBW goat kids were obtained using the Wilcoxon rank-sum test.

**Table 4 animals-14-00425-t004:** Comparison of the relative abundance of the rumen microbiota at the genus level (relative abundances > 0.10% in at least 70% of samples) of weaning goat kids.

Phylum	Genus	Group ^1^		SEM ^2^	*p*-Value ^3^
		HBW	LBW		
Bacteroidetes	*Bacteroides*	3.03	10.28	3.62	0.01
	*Bacteroidales_UCG-001*	1.31	0.01	0.45	0.05
	*F082*	19.61	1.35	9.15	0.03
	*Butyricimonas*	0.15	1.61	0.72	0.05
	*Prevotellaceae_UCG-001*	4.88	0.13	2.37	0.01
	*Prevotellaceae_UCG-003*	3.32	0.04	1.35	0.05
	*Prevotellaceae_UCG-004*	0.83	0.03	0.25	0.02
	*Prevotella*	0.45	0.23	0.14	0.04
	*Alistipes*	0.76	3.74	1.49	0.05
	*Rikenellaceae_RC9_gut_group*	9.70	1.85	3.92	0.05
Firmicutes	*Clostridium_sensu_stricto_1*	0.18	1.20	0.51	0.05
	*Ruminococcus_torques_group*	0.23	3.01	1.39	0.01
	*Monoglobus*	2.52	4.42	0.50	0.03
	*Oscillospiraceae_UCG-005*	1.49	7.00	2.75	0.05
	*Ruminococcus*	1.92	0.07	0.55	0.02
	*Pseudobutyrivibrio*	0.43	0.01	0.14	0.05
	*Defluviitaleaceae_UCG-011*	0.12	0.01	0.03	0.04
	*Veillonellaceae_UCG-001*	0.37	0.01	0.13	0.03
	*Eubacterium_nodatum_group*	0.27	0.01	0.10	0.04
	*Lachnospiraceae_UCG-002*	0.23	0.01	0.02	0.05
	*Syntrophococcus*	0.17	0.08	0.06	0.02
Proteobacteria	*Escherichia-Shigella*	1.40	17.47	8.03	0.03

^1^ HBW = high body weight; LBW = low body weight. ^2^ SEM, standard error of the mean. ^3^ Adjusted *p*-value = false discovery rate-adjusted *p*-value; the *p*-values between the HBW and LBW goat kids were obtained using the Wilcoxon rank-sum test.

## Data Availability

The raw datasets analyzed in this study are accessible in the SRA (Sequence Read Archive) of the NCBI (National Center for Biotechnology Information) database. These datasets are cataloged under the specific Bio project accession number PRJNA978501, facilitating easy retrieval and review for interested parties.

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
