# Peer review of "Exploring the Rumen Microbiota and Serum Metabolite Profile of Hainan Black Goats with Different Body Weights before Weaning"

_animals, 2024, doi:10.3390/ani14030425_

Round 1
Reviewer 1 Report
Comments and Suggestions for Authors
General comments:
This is a pretty interesting study that particularly focused on the goats' weaning body weight and its potential association with rumen microbiota and serum metabolites. The identified key bacterial genera and metabolites may help humans better understand the difference between HBW and LBW goats, as well as the potential targeting of probiotics and/or prebiotics.
The manuscript is well written and the story is clear and meaningful. It’s also in the scope of the Journal of Animals and can highly advance the knowledge in goat science. I suggested it can be accepted by the Journal of Animals.
However, there are still several concerns that should be revised before the acceptance.
1. In the abstract, please don’t include unnecessary abbreviations. Especially for those only showing up for once.
2. In Table 1, what is the SEM, please give the annotation of the SEM.
3. In Table 2, What are the NH3-N and TVFA, please clarify.
4. In line 321, please double-check the Spearman |r| or r square.
5. In Figure 3, please enlarge the font size, it is really difficult to see the word in the figure
6. In line 486, it’s not strain. In your study, you use amplicon sequence data, it is unable to identify the bacteria strain. Bacterial genera or you can say microbial taxa, please revise it.
7. In line 488, what do you mean elevated level? Is it a higher concentration or higher proportion? Please revise it.
8. In line 488, what’s the difference between the more “effective” and “efficient”? If possible, just be brief.
9. There are some inconsistencies for the P or P, italic or not? Please double-check throughout the manuscript carefully.
10. The figures in the manuscript was not clear. Please use the clear figures.
Author Response
Thank you very much for your guidance and suggestions. The following is the response to the questions raised and corresponding modifications have been made in the article.
1: The unnecessary abbreviations in the abstract have been removed.
2: SEM, Standard Error of the Mean: abbreviated as SE, standard error, is also the standard deviation (how the mean is distributed) of the estimated sample mean. The standard error is used to measure sampling error. A smaller standard error indicates that the sample statistics are closer to the values of the population parameters, and the sample is more representative of the population. The reliability of inferring population parameters using sample statistics is greater.
3: Total volatile fatty acids (TVFA), acetic acid, propionic acid, butyric acid, valeric acid, isobutyric acid, and isovaleric acid concentrations in the rumen fluid of lambs. The concentration of ammonia nitrogen (NH3-N) in the rumen reflects the relationship between the production rate and utilization rate of ammonia nitrogen in the rumen and is an important indicator for evaluating N utilization efficiency
4: Spearman | r | or r squared has been modified to r>0.6, r<-0.6.
5: The text in Figure 3 has been bolded, and the image has been enlarged.
6: The word 'strain' has been replaced with 'microbial taxa'.
7, 8: Content to be modified
9: P is italicized.
10: Modified.
Reviewer 2 Report
Comments and Suggestions for Authors
The authors investigated the impact of the microbiome on the weaning weight of goats by analyzing metabolites in both rumen and serum. This paper is well-conceived, meticulously executed, and effectively presented. However, there are a few minor points that should be addressed:
The resolution of all figures is too low. Even when zoomed in, they remain unclear. Please enhance the resolution.
In line 63, consider using the minor form of "we." Therefore, we hypothesized...
In line 241, the text mentions an average of 2,588 OUT, while Table S4 presents 4162 and 2905 OTUs in HBW and LBW, respectively. Please review and correct for consistency.
In line 322, "R2Y" and "Q2" should be correctly formatted as R2Y and Q2.
In line 396, consider using "transfaunation" instead of "translocation."
Comments on the Quality of English Language.
Author Response
Thank you very much for your feedback and suggestions. Based on your suggestions, this article has made revisions to the images and their main text.
Reviewer 3 Report
Comments and Suggestions for Authors
General comment
The hypothesis of the study needs to be changed, the analyzes and results presented do not allow for its evaluation, and the evaluation of the hypothesis itself was not presented in the discussion.
The title and objective of the study must be changed, the analyzes and results presented in the study only allow the comparison between the rumen microbiota and blood metabolite profile between "kids or goats" with different growth rates before weaning.
Abstract
The abstract mentions the study with lambs but the title describes goats, which species was used? describe correctly throughout the text.
In the abstract it is necessary to present the most important results and the conclusion of the study
Introduction
In general, the introduction does not make it clear what the objective of the study is.
Line 66-68: this is not part of the introduction, this is part of the material and methods and does not add to the introduction.
line 68-71 replaces this paragraph with the objective of the study
M&M
It is necessary to replace the term lamb with kids throughout the text, sheep's offspring are lambs and goat offspring are kids, as it is written there is doubt as to which animal species was used.
Results
The quality of figures 2 and 3 A-B needs to be improved
Discussion
The authors need to understand that there was no meticulous evaluation of the microbial profile, in fact the results demonstrate a superficial analysis of the microbial profile.
The discussion needs to be rewritten in order to explore the results of the study and not just an excessive comparison with the results of other studies.
Line 380-382: Present this information about ruminal morphology: If it does not exist, delete it from the text.
Conclusion
The conclusion must be rewritten to present the main results of the study, in its current form and just a continuation of the discussion.
Author Response
Thank you very much for your guidance and suggestions. The following is the response to the questions raised and corresponding modifications have been made to the article.
Corresponding revisions have been made to the title, assumptions, purpose, and discussion section of the article.
Abstract
Change the name of the experimental object to goat kids. The abstract has been revised and the research results and conclusions have been summarized. To highlight the relationship between the before weaning period and the Hainan black goat species in the title, the term goat kids is used as a collective term in the main text of the manuscript, which is not contradictory.
Introduction
The introduction section has been revised and the research purpose has been explained.
M&M
Replace the word "lamb" with "goat kids" throughout the text.
Results
The images in the main text have been enlarged and the text in the images has been bolded.
Discussion
The discussion section has been revised, and most of the literature references use recent research articles as support for the argument.
Conclusion
The conclusion section summarizes and re-discusses the experimental results.
Finally, thank you once again for your suggestions and constructive feedback. Based on your suggestions, the manuscript has been revised accordingly.
Reviewer 4 Report
Comments and Suggestions for Authors
The animals-28232109 clarifies the relationship between differences in weaning weight of goat kids and the rumen microbiota. Although the purpose of the experiment was interesting, I thought the methodology and results needed consideration.
Genetic differences and epigenetic physiological differences are attracting attention as factors contributing to differences in the growth of young goats. On the other hand, I thought it was interesting and challenging that the authors investigated the rumen microbiota using kid goats with the same genetic background. However, the rumen microbiota is influenced by feed intake. Therefore, I thought that information on the respective feed intakes of HBW and LBW is necessary. The results of this paper would be a major discovery if the growth of young goats were different just because of the difference in rumen microbiota even if the amount of feed intake was the same.
There are also some problems with this paper. It is difficult to understand the relationship between the figure and the text.
What is the X axis in Figure 1?
Is Figure 2 appropriate to show the L300-312 in the main text? Many items in Figure 2 are interrelated, and it is difficult to understand which relationships are important in this type of firgure.
I thought that the impact numbers in Figure 3 and L333-337in the main text did not match.
Please check if this paper will help you with your research. https://doi.org/10.1016/j.jprot.2023.104982
Author Response
Thank you very much for your guidance and suggestions. The following is the response to the questions raised and corresponding modifications have been made to the article.
Firstly, we agree with your point of view. Currently, research reports have shown that genetic and epigenetic differences can indeed cause differences in the growth of young goats. We have chosen a large-scale Hainan black goat breeding plant that has not undergone breed improvement as our experimental base to minimize the influence of genetic factors on the experimental results. Secondly, dietary factors are one of the important factors affecting animal growth. During the experiment, the young goats were regularly and quantitatively fed to avoid bias in the experimental results due to differences in dietary formula and intake. Of course, it's just an experiment and not perfect. Throughout the entire experiment, we tried our best to minimize external interference in the test results. Therefore, based on your suggestions, the entire text has been revised and described in more accurate language.
The following four are responses to your questions.
1: The difference in the x-axis can explain 25.5% of the comprehensive analysis results. The dots of different colors represent the sample groups of rumen microbiota of different goat kids with different body weights. The scale on the horizontal and vertical axes represents relative distance and has no practical significance.
2: To clarify the specific microbial community for in-depth research, the correlation between rumen microbiota and measurement indicators in Figure 2 was studied, the measurement indicators were screened and the shared microbial community was analyzed with the serum metabolome in Figure 3.
3: Figure 3 has been corrected with the text and numbers in the main text.
4: The recommended paper has been cited as the 54th reference in the manuscript.
Finally, thank you once again for your suggestions and constructive feedback. Based on your suggestions, the manuscript has been revised accordingly.
Round 2
Reviewer 3 Report
Comments and Suggestions for Authors
It is still necessary to improve the quality (resolution) of the images
Author Response
Thank you for your suggestion. The manuscript has already included images that adhere to the journal's specifications in terms of size and clarity.

Reviewer 4 Report
Comments and Suggestions for Authors
It is very unfortunate that DMI results do not exist. On the other hand, it is noteworthy that there was a large difference in body weight gain even when the feed intake was controlled so that there was no difference. Therefore, I recommend stating in the text that the experimental results are not biased by differences in dietary intake, since the goat kids ate the same amount of food.
Also, I don't understand Figure 3. Which part of Figure 3 did you modify? Although impact is written on the X-axis of Figure 3B, only three items are greater than 0.1. Please check again if it matches the L340-344.
Author Response
Thank you very much for your suggestions and for adding appropriate content. We had a profound discussion before the experiment and realized that dry matter intake (DMI) during the experiment was an important factor affecting the weight of goat kids. Therefore, in the preliminary experiment, we measured the DMI of goat kids in the goat factory. During the experiment, ensure that there is no leftover feed for the daily goat kids and meet the nutritional requirements (concentrate: 2.0% of body weight, coarse: 3.0% of body weight). To avoid any remaining feed in the LBW group, the feed intake on the day before the end of the experiment was 220g, which was lower than the calculated value of 278g. In such similar experiments, the DMI of high body weight recombination is usually higher than that of low body weight recombination, and the effect on body weight is explained by calculating the DMI. Possible biases in experimental results may be caused by DMI overlapping factors, as listed in the latest published paper "Preweaning period is a critical window for rumen microbial regulation of average daily gain in Holstein heifer calves" (https://jasbsci.biomedcentral.com/articles/10.1186/s40104-023-00934-0#Sec16). It is estimated that Holstein heifer calves have an average daily increase of 30-50g in DMI, and the accumulated DMI can reach 2700-4500g over a 90-day experimental period, which can affect body weight and rumen microbiota.
The new image has been uploaded again.
Thank you again for your questions and suggestions.